# High-resolution cryo-EM structure of the *Pseudomonas* bacteriophage E217

Fenglin Li [1,3], Chun-Feng David Hou [1,3], Ravi K. Lokareddy[1], Ruoyu Yang[1], Francesca Forti[2], Federica Briani [2] ✉ & Gino Cingolani [1] ✉

E217 is a *Pseudomonas* phage used in an experimental cocktail to eradicate cystic fibrosis-associated *Pseudomonas aeruginosa*. Here, we describe the structure of the whole E217 virion before and after DNA ejection at 3.1 Å and 4.5 Å resolution, respectively, determined using cryogenic electron microscopy (cryo-EM). We identify and build de novo structures for 19 unique E217 gene products, resolve the tail genome-ejection machine in both extended and contracted states, and decipher the complete architecture of the baseplate formed by 66 polypeptide chains. We also determine that E217 recognizes the host O-antigen as a receptor, and we resolve the N-terminal portion of the O-antigen-binding tail fiber. We propose that E217 design principles presented in this paper are conserved across PB1-like *Myoviridae* phages of the Pbunavirus genus that encode a ~1.4 MDa baseplate, dramatically smaller than the coliphage T4.

Bacteriophage E217 infects *Pseudomonas aeruginosa* and is part of an experimental phage cocktail developed to eradicate *P. aeruginosa* infections in *Galleria mellonella* (wax moth) larvae and vertebrate models[1-3]. E217 is a PB1-like *Myoviridae* of the Pbunavirus genus characterized by a ~66.2 kbp genome[1], significantly smaller than the classical enterobacteria phage T4 (~169 kbp)[4]. PB1-like phages are ubiquitous on Earth[5], found in freshwater in the US[6] and Brazil[7], as well as sewage and wastewater in Europe[8]. These phages have genomes of ~65 kbs and significantly simpler baseplate complexes than the classical coliphage T4. A unique fivefold vertex of the E217 capsid is occupied by a long, contractile tail apparatus that breaks the icosahedral symmetry[9]. The tail assembles onto a dodecameric portal protein that replaces a capsid penton at the unique vertex, providing an entry and exit channel for the virus to package and eject DNA[10]. The E217 genome is packaged into an immature precursor capsid (or procapsid) using two terminase subunits, TerL and TerS that we recently characterized[11]. Concurrently, the baseplate assembles and recruits the tail tube, sheath proteins, and fibers attaching to the portal vertex through a neck formed by additional factors.

Most current knowledge about *Myoviridae* assembly is extrapolated from the model system T4, which has been studied extensively for over one century[12-14]. In *Myoviridae*, the baseplate is a multisubunit complex that harbors different activities, including attachment to the host surface, penetration of the host inner membrane, and contraction-coupled genome ejection. The architecture of the isolated T4 baseplate has been elucidated in the pre- and post-contraction states, taking advantage of a mutant (T4-18am-23am) that assembles the entire baseplate-tail tube complex but fails to incorporate it into a virion[15,16]. The T4 baseplate is a ~6 MDa machine built by 15 different polypeptide chains repeated in several copies to generate a ~145 proteins molecular complex. The T4 baseplate is ~490 Å in diameter, twice that of the bacterial ribosome, and contains both short- and long-tail fibers responsible for phage attachment to the bacterial cell wall[17,18]. Hu et al. analyzed T4 at various stages of infection using cryo-electron tomography (cryo-ET)[19]. Interestingly, most long-tail fibers are folded back against the virion before infection and do not interact directly with the host surface. The first event of tail contraction is the irreversible binding of short tail fibers protruding from the bottom of the baseplate to the host outer membrane, which triggers sheath contraction. This event drives the tail tube into the periplasm with subsequent genome ejection. Host membrane puncturing does not

[1]Department of Biochemistry and Molecular Biology, Thomas Jefferson University, 1020 Locust Street, Philadelphia, PA 19107, USA. [2]Dipartimento di Bioscienze, Università degli Studi di Milano, Milan, Italy. [3]These authors contributed equally: Fenglin Li, Chun-Feng David Hou. ✉e-mail: federica.briani@unimi.it; gino.cingolani@jefferson.edu

require the proton motive force, which only becomes necessary for genome translocation.

A medium-resolution cryo-EM analysis of the Twort-like Myoviridae phage phi812 was reported[20]. This bacteriophage contains a baseplate drastically different than T4, with baseplate proteins organized into two layers parallel to the bacterial cell wall. A similar double-layered baseplate was also found in the *Staphylococcus aureus* phage SaGU1[21]. More recently, the atomic structure of the R2 pyocin was determined in the pre- and post-contraction states using cryo-EM single particle analysis (SPA)[22–24]. R-type pyocins are minimal contractile nanomachines, similar to Myoviridae tails, but lacking a phage head and genetic information. R-type pyocins share structural similarities with the Photorhabdus virulence cassette (PVC) contractile injection systems, which, unlike pyocins, can target eukaryotic cells[25]. The cryo-EM structure of the R2 pyocin in the pre- and post-contraction states consists of 11 unique factors repeated in hundreds of copies, of which eight proteins form the baseplate. The R2 pyocin baseplate is minimal compared to the T4 assembly but exerts the same essential function of host attachment and signal transduction to initiate tail contraction and membrane penetration. The pyocin work suggested that the tail fibers are connected laterally to the baseplate and upon receptor-binding, initiate a cascade of events that lead to sheath contraction[22]. Fraser et al. elaborated on this idea[26] and found that the contractile tail has an activation energy of 160 kcal/mol and develops a force greater than 500 pNs during contraction, providing the energy necessary to penetrate membranes. The contraction mechanisms elucidated for the R2 pyocins likely apply to other contractile injection systems, such as type VI secretion systems (T6SS) and phage tails[27].

More recently, a high-resolution cryo-EM reconstruction of the freshwater *Myoviridae* cyanophage Pam3 revealed the architecture of a minimal *Myoviridae* baseplate formed by only six unique polypeptide chains[28]. Strikingly, Pam3 baseplate is covalently cross-linked via disulfide bonds to twelve tail fibers that alternate in an upward and downward configuration. In this work, we describe the high-resolution structure of the *Pseudomonas* phage E217, which we visualize before and after tail contraction using cryo-EM. The structures presented here shed light on the conformational dynamics of a small *Myoviridae* used in an experimental phage cocktail, elucidating the proteins and mechanisms of tail contraction-induced genome ejection.

## Results

### Cryo-EM reconstruction of E217 with the tail in the extended and contracted conformation

We used cryo-EM SPA to visualize the E217 icosahedral capsid and tail apparatus (Fig. 1a). Aiming at a high-resolution reconstruction of the tail in the extended and contracted states, we collected two large datasets (~44,000 movies) of the E217 mature virion at physiological and basic pH, respectively (Table 1). At physiological pH, most E217

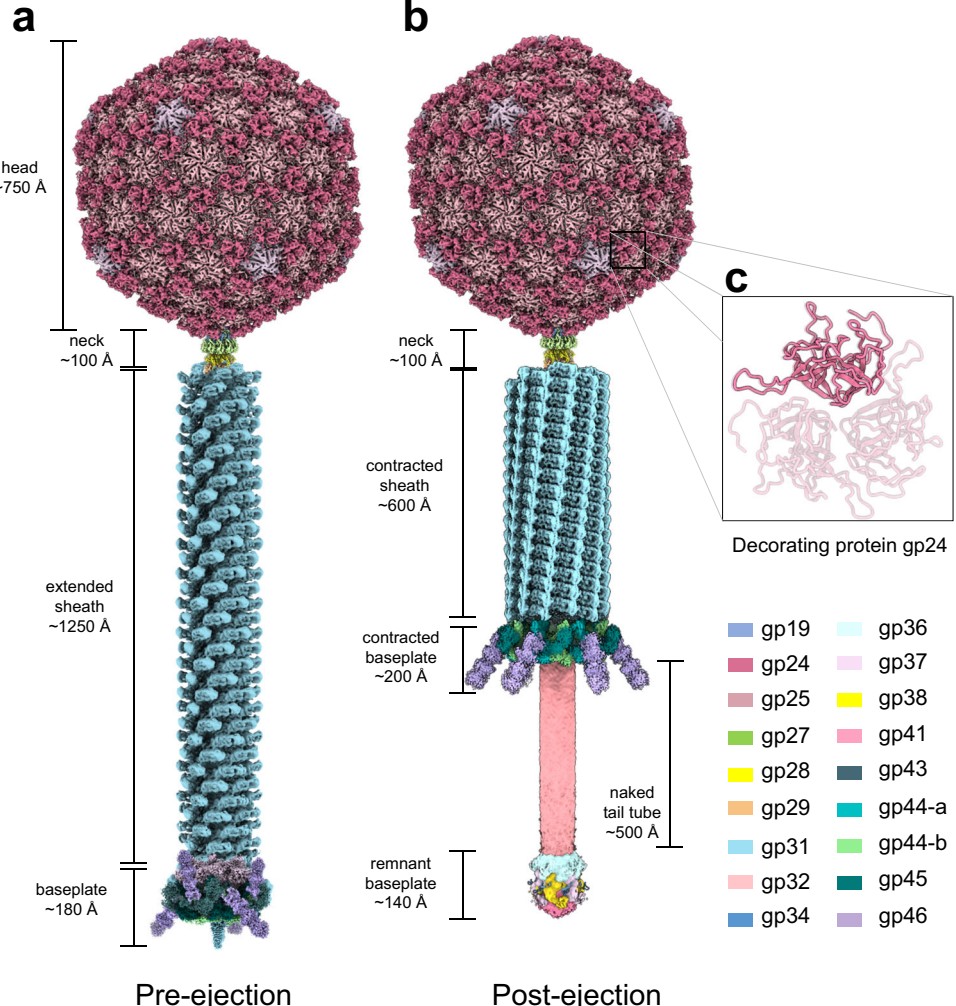

**Fig. 1 | Global view of phage E217 in pre- and post-ejection conformation.** Composite cryo-EM reconstructions of the E217 virion at neutral pH (**a**) and after incubation at alkaline pH (**b**). **c** Zoom-in view of the trimeric cementing protein gp25 that decorates the icosahedral capsid.

**Table 1 | Map and model refinement statistics**

| | Bacteriophage E217 extended tail | | Bacteriophage E217 contracted tail | | |
|---|---|---|---|---|---|
| **Data collection and processing** | | | | | |
| Facility/microscope | NCEF/Titan Krios | | NCCAT/Titan Krios | | |
| Camera | Gatan K3 | | Gatan K3 | | |
| Magnification | ×81,000 | | | | |
| Voltage (kV) | 300 | | | | |
| Electron exposure (e⁻/Å²) | 50 | | | | |
| Defocus range (μm) | −0.75 to −1.5 (step 0.25) | | | | |
| Pixel size (Å) | 1.12 | | 1.058 | | |
| Total movies (frames/movie) | 22,015 (40) | | 22,714 (40) | | |
| **Refinement** | | | | | |
| PDB entry code | 8FRS | 8FVH | 8FUV | 8FVG | 8EON |
| Particles per reconstruction | 13,302 | 15,505 | 13,257 | 10,826 | 10,126 |
| Symmetry | C5 | C6 | C6 | C6 | C3 |
| Model name | capsid: decorating | portal:H-to-T: collar:gateway | extended sheath:tube | contracted sheath | baseplate components |
| Map resolution (Å) | 2.8 | 3.4 | 3.1 | 3.1 | 3.6 |
| FSC threshold | 0.143 | | | | |
| Initial model used | De novo | De novo | De novo | De novo | De novo + AlphaFold |
| Correlation coefficient (CC) | 0.88 | 0.84 | 0.80 | 0.80 | 0.70 |
| Model composition | | | | | |
| Number of chains | 14 | 36 | 7 | 13 | 45 |
| Non-hydrogen atoms | 31,497 | 70,392 | 11,202 | 48,022 | 95,688 |
| Residues | 4120 | 8886 | 1500 | 6409 | 12,711 |
| RMS deviations | | | | | |
| Bond lengths (Å) | 0.002 (0) | 0.003 (0) | 0.003 (0) | 0.003 (0) | 0.009 (21) |
| Bond angles (°) | 0.5 (2) | 0.6 (22) | 0.7 (3) | 0.5 (9) | 0.7 (74) |
| Validation | | | | | |
| MolProbity score | 1.73 | 1.71 | 2.08 | 1.71 | 2.19 |
| Clashscore | 5.10 | 7.26 | 5.09 | 5.94 | 13.20 |
| Rotamer outliers (%) | 0.00 | 0.07 | 0.00 | 0.04 | 0.02 |
| Ramachandran plot | | | | | |
| Favored (%) | 92.60 | 95.5 | 93.52 | 94.32 | 89.52 |
| Allowed (%) | 7.1 | 4.3 | 6.3 | 5.6 | 10.1 |
| Outliers (%) | 0.29 | 0.15 | 0.00 | 0.00 | 0.36 |

virions had an extended tail about 1500 Å long, with density for DNA inside the head. The tail, divided into a neck, sheath, and baseplate, had excellent density for all components. In contrast, basic pH triggered irreversible tail contraction and genome ejection[22] (Supplementary Fig. 1a). We reconstructed the E217 head using a localized reconstruction with fivefold symmetry imposed (C5), which yielded 2.8 Å resolution at 0.143 cut-off Fourier Shell Correlation (FSC). We also visualized all components that deviate from icosahedral symmetry[9] using localized reconstruction[29] (Supplementary Fig. 1b) and local symmetry averaging, which yielded 3.3 Å resolution for the neck, 3.1 Å for the sheath and 3.4 Å for the baseplate (Supplementary Fig. 1c), at FSC₀.₁₄₃. The high-quality electron densities allowed us to build de novo models of all components in the phage neck, sheath, and baseplate, including the most distal region of the tail fibers. In total, we built 19 new polypeptide chains (Supplementary Fig. 2), which were real-space refined to a final Correlation Coefficient (CC) of ~0.70 to 0.88, indicating an excellent model-to-map fit (Table 1). Treatment with alkaline pH triggered in vitro sheath contraction[22], exposing ~500 Å of the tail tube (Fig. 1b), although the resolution of the contracted empty particles devoid of DNA was limited to 4.5 Å. Figure 1a, b shows composite reconstructions of the E217 with the extended and contracted tail representing the pre- and post-ejection conformation obtained by combining different cryo-EM maps (e.g., C5

capsid: EMD-29406; C6 neck: EMD-29487; C6 extended: EMD-29481; C6 contracted sheath: EMD-29486; C3 baseplate: EMD-28405).

## Gp24 decorates the E217 capsid

The E217 head has a maximum diameter of 790 Å and is built by 535 copies of the major capsid protein gp25. We will refer to the reconstruction of the E217 virion at neutral pH with an extended tail (Fig. 1a) that was determined at an FSC = 0.143 resolution of 2.8 Å (Supplementary Fig. 1c), although no significant differences were observed in the E217 head because of alkalization. The E217 major capsid protein, gp25, adopts a canonical HK97 fold[30] (Supplementary Fig. 3a–c) and is organized into a hexagonal lattice with a triangulation number $T = 9$[31]. The first 65 amino acids of gp25 were not visible in the reconstruction, suggesting this moiety could be cleaved during maturation or is disordered in our reconstruction. The rest of gp25 (res. 66–382) adopts an HK97 fold that a DALI search[32] identified as being most similar to the major capsid protein from the marine *Siphovirus* TW1 (PDB 5WK1 [10.2210/pdb5wk1/pdb])[33], which is only 10% identical to E217 gp25 in amino acid sequence. The RMSD between E217 and TW1 capsid proteins is 8.8 Å for 274 aligned Cα atoms out of 296 and 317 Cαs, respectively (Supplementary Fig. 3d).

 The E217 icosahedral head is decorated by 180 copies of a trimeric protein, gp24 (res. 1–211), that binds at the icosahedral threefold and

quasi-threefold axes (Fig. 1a, b), generating a surface protrusion. Gp24 adopts a β-tulip fold[34] (Fig. 1c), like the decorating protein gp87 found in the thermophilic phage P74-26[35]. Each gp24 subunit consists of two anti-parallel β-sheets, arranged like the petals of a flower, to form a tulip-like structure. An extended N-terminal arm (res. 1–25) emanates from each subunit and packs laterally with an omega loop connecting residues 58–75 from an adjacent subunit (Supplementary Fig. 4a). Each E217 gp24 trimer contacts three neighboring decorating proteins and six capsid protein subunits in three adjacent capsomers, burying a total surface area of ~8642 Å². Gp24 trimers at neighboring threefold/quasi-threefold axes generate an inter-capsomer cage (Supplementary Fig. 4b). The interconnected architecture of gp24 trimers is stabilized by extensive van der Waals contacts between N-terminal arms and omega loops.

## The E217 neck
One unique penton of the E217 capsid is replaced by the dodecameric portal protein gp19, which is an attachment point for the ~1500 Å tail apparatus. The first part of the E217 tail, or neck, was visualized in a localized reconstruction after applying C12 or C6 rotational averaging, which yielded maps at 3.3 and 3.4 Å resolution (FSC = 0.143), respectively (Supplementary Fig. 1c). The C12 map was used to build de novo models of the dodecameric portal protein gp19 (res. 97–528 out of 765), and head-to-tail adaptor gp27 (res. 1–155) (Fig. 2a and Supplementary Fig. 2). The density under the head-to-tail factor was better resolved in a C6 map, consistent with a reduction of rotational symmetry. We identified and built six copies of the collar protein gp28 (res. 1–132) and gateway gp29 (res. 1–183) (Fig. 2a, b and Supplementary Fig. 2).

E217 portal protein shares the general portal protein fold[10]. However, unexpected for a *Myoviridae*, the E217 portal also contains a C-terminal α-helical barrel projecting into the head (Fig. 2a). The barrel domain has been predicted[36] and experimentally observed in *Podoviridae* phages like P22[36–38] and Sf6[39] but not found in *Myoviridae*[40]. Interestingly, the barrel is folded only in the pre-ejection conformation when DNA fills the capsid (Fig. 2a). The pre-ejection reconstruction also revealed a continuous density threading through the portal protein and neck channel (Fig. 2a), consistent with double-stranded DNA (dsDNA). In contrast, treatment with alkaline pH triggered E217 tail contraction and genome ejection. Unexpectedly, the portal protein barrel became unstructured in the post-ejection state due to the lack of bonding interactions with dsDNA (Fig. 2b), as in the P22 procapsid[38]. E217 portal dodecamer is attached to twelve copies of the head-to-tail factor gp27, which closely resembles P22 gp4[37]. Gp27 assembles to the portal by inserting its extended C-terminal tail at the interface generated by two portal subunits, suggesting the head-to-tail factor oligomerizes upon binding instead of existing as a pre-formed dodecameric ring[41].

The other two neck factors, gp28 and gp29, reduce the C12 symmetry found in the portal and head-to-tail to sixfold, which is maintained in most of the tail apparatus. Gp28 consists of a small β-barrel decorated by a prominent N-terminal α-helix and two lateral extensions (res. 28–38 and 56–66) (Supplementary Fig. 5a) that give it the appearance of a horse saddle. Six collar proteins pack under the head-to-tail dodecamer with gp28 N-helix positioned orthogonal to two gp27 subunits, explaining the C12:C6 symmetry mismatch (Supplementary Fig. 5b). On the opposite side of the gp28 barrel, the wings contact two gateway subunits like a saddle sitting onto a horse (Supplementary Fig. 5b).

The last neck factor, gp29, connects the neck to the sheath lattice (Fig. 2a, b). Gp29 shares a 10% sequence identity with the tail terminator protein from phage lambda (gpU-D74A), which is also hexameric[42]. It makes two sets of contacts: on one end, it sits onto the hexameric tube built by the tail tube protein gp32, generating a continuous β-sheet ridge that delimits the tail lumen (described below). On the other hand, six gp29 C-terminal extensions (res. 172–182) extend outward to recruit six sheath proteins positioned laterally to the gateway (Fig. 2c). Collar and sheath proteins associate together by forming an intermolecular 3-stranded β-sheet. This secondary structure element is formed by a C-terminal β-turn in the sheath protein (β-strands #1 and #2, res. 460–488) (Fig. 2c) and gp29 C-terminal extension (β-strand #3, res. 172–182). The intermolecular β-sheet bridging collar and sheath proteins are maintained in the contracted state, where the gp29 C-terminal arm swings by ~25° to accommodate an increase in the sheath diameter to 250 Å (Fig. 2d).

## E217 sheath quaternary structure
The E217 sheath consists of 204 copies of the tail sheath protein gp31 (res. 1–204)[43] that surrounds 34 staked hexameric rings of the tail tube protein gp32. We determined cryo-EM reconstructions of the extended and contracted E217 sheath at 3.1 Å resolution (FSC = 0.143) (Supplementary Fig. 1c). Tertiary and quaternary structure changes in the sheath protein gp31 cause sheath contraction. At the quaternary structure level, the extended conformation of the E217 sheath consists of a helical tube of ~205 Å in diameter that has a regular pitch of 490 Å and rise and twist of 40.8 Å and 31.3°, respectively (Fig. 3a). The extended sheath is porous, with visible openings between subunits. The internal lumen is ~75 Å and houses the tail tube (described in the next section). Alkalization or spontaneous contraction during purification, observed for ~20% of all virions at neutral pH, triggers tail contraction. This results in sheath compression, with the pitch decreasing to 265 Å and the outer diameter increasing to 250 Å, while the rise and twist decrease to 22.1 Å and 30°, respectively (Fig. 3b). A superimposition of the extended versus contracted states reveals the remarkable degree of sheath compression that loses 55% of its length to increase in diameter by 20% (Fig. 3c).

## E217 sheath protein conformational dynamics
The sheath protein gp31 contains three globular domains, referred to as A, B, and C, and long N- and C-terminal extension arms (res. 1–23 and 491–504) emanating from domains B and C, respectively (Supplementary Fig. 6a). E217 domains B and C have substantial structural similarity to the R2 pyocin sheath protein[22], while domain A, missing in the smaller pyocin sheath[22], is conserved in the T4 sheath protein (Supplementary Fig. 6a). Despite the similar domain composition and overall topology, the E217 and T4[43] sheath proteins are vastly different in amino acid sequence and structure (RMSD ~26 Å with only 5 aligned Cα atoms out of 504 and 479 Cαs, respectively) (Supplementary Fig. 6b).

A secondary structure superimposition of the E217 sheath protein in the extended versus contracted conformations (Fig. 3d) revealed that the N-terminal arms undergo significant movement, with an 80° rotation around residue 24 upon contraction and a maximum displacement of up to 20 Å. A smaller movement occurs for the C-terminal arm, where residues 491–504 rotate by ~12° and the β-turn (res. 460–488) (Fig. 2c, d) shifts by ~8 Å. Interestingly, these three moieties (Fig. 3d) dictate the assembly of sheath subunits into a lattice. In the extended sheath, the three moieties come together from neighboring subunits to form an intermolecular four-stranded β-sheet (Fig. 4a). Specifically, a sheath subunit A exposes the C-terminal β-turn (res. 460–488) that contains β-strands #1 and #2. The N-terminal arm of a neighboring sheath subunit B located above the plane of subunit A provides β-strand #3, and β-strand #4 originates from the C-terminal arm of a neighboring subunit, located slightly above subunit A but rotated clockwise. Notably, this fourth strand is shorter, with only six amino acids (res. 6–12) engaging in main chain hydrogen bonding and has a sharp 90° bent at Gly14. Sheath contraction results in a dramatic quaternary structure reorganization with a ~17° clockwise rotation of the β-sheet and extension of the β-strand #4 that becomes nearly

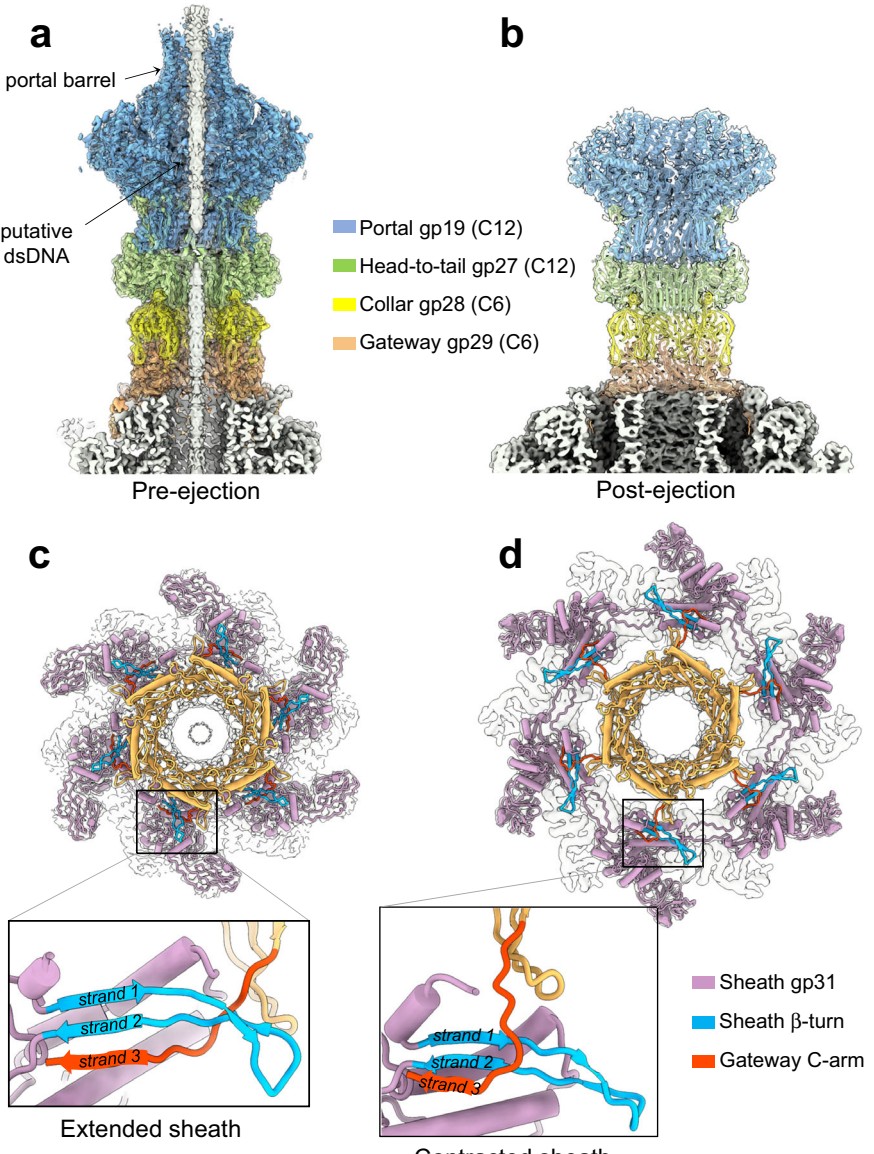

**Fig. 2 | Architecture of the E217 neck before and after genome ejection.** Section views of the E217 neck area in pre- (**a**) and post-ejection (**b**) conformations. Refined atomic models are overlaid with the cryo-EM density displayed at 3.5 σ contour above the background. **c**, **d** Top views of the gateway hexamer (orange) bound to six sheath proteins (purple) in the pre- / post-ejection conformations. The zoom-in panels reveal a detail of the gateway:sheath association. The gateway C-terminal β-strand #3 (colored in red) forms an intermolecular β-sheet with the sheath protein C-terminal β-turn (strands #1–2, colored in blue).

straight C-terminal of Gly14 (Fig. 4a). This results in striking compaction and contraction of the sheath that expands in diameter (Fig. 4b, c), exposing ~500 Å of the naked tail tube (Fig. 1b).

**E217 tail tube**

Stacked hexamers of the tail tube protein gp32 form the central tube of the E217 tail. The tail tube protein is structurally conserved and homologous to the tail tube protein found in non-contractile tails, bacterial pyocins, and type VI secretion systems[44]. Gp32 consists of a classical β-sandwich-type fold that contains two 4-stranded sheets facing each other and surrounded by a long α-helix (res. 67–79) and an extended β-turn (res. 35–56) (Supplementary Fig. 7a). Six copies of gp32 assemble laterally to form a hollow hexamer, whose internal lumen measures ~40 Å in diameter. The interface between gp32 subunits is stabilized by 11 hydrogen bonds, many of which form between residues exposed by the extended β-turn. However, the gp32:gp32 interface lacks hydrophobic contacts, suggesting the tail tube protein

may remain monomeric in solution without other tail components[45]. The interior surface of the tube is mildly charged, with two charged patches, one positive, exposed by Arg125/Lys127, and the other negative, formed by Asp27 (Supplementary Fig. 7b).

The interface between the tail tube and sheath protein varies depending on the state of tail contraction. Before genome ejection, each tail tube subunit contacts one sheath protein generating an extensive interaction surface stabilized by two hydrogen bonds, one salt bridge, and 99 van der Waals contacts (Supplementary Fig. 7c). Notably, tail sheath subunits do not contact each other in the extended tail but are solely connected by the association of their C-terminal β-turns and N-/C-terminal arms (Fig. 4a, b). Upon contraction, tail sheath bodies come into contact laterally, forming a much tighter lattice that releases a large amount of energy[26]. Because of the increased diameter in the contracted state (Fig. 4c), sheath subunits do not make direct bonds with the tail tube, whose only point of contact is with the gateway C-terminal extension (Fig. 2d).

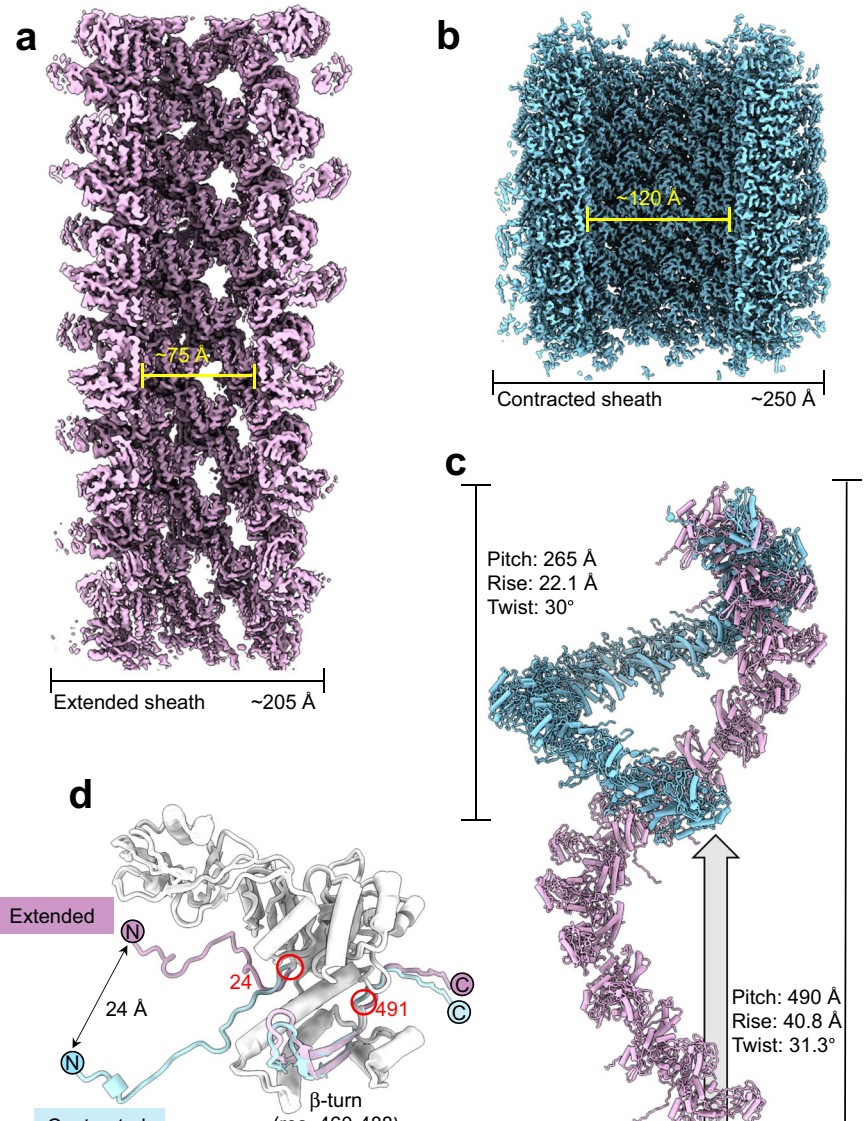

**Fig. 3 | Atomic structure of the E217 sheath.** Cryo-EM reconstructions of the contracted (**a**) and extended (**b**) sheaths calculated at 3.1 Å resolution at 0.143 FSC and displayed at 3 σ. **c** Ribbon diagram of all sheath subunits in one helical pitch of extended (purple) versus contracted (cyan) sheaths. **d** Superimposition of sheath protomers in the extended versus contracted conformations. Three regions undergo the most significant conformational rearrangements: the N-terminal arm (res. 1–23), C-terminal β-turn (res. 460–488), and C-terminal tail (res. 491–504).

## De novo structure of the E217 extended baseplate at 3.4 Å resolution

We determined a localized reconstruction of the extended baseplate at 3.4 Å resolution (FSC = 0.143) (Fig. 1a) and a lower resolution reconstruction after contraction (Fig. 1b). We built a complete atomic model of all factors in the extended baseplate, which consists of 11 unique proteins repeated in 66 copies. The total mass of the baseplate is 1.4 MDa excluding six trimeric tail fibers emanating at the wedges of the baseplate. The baseplate assembles at the tip of the tail apparatus, breaking the repetitive symmetry of the tail tube and sheath protein, creating a unique tip for host attachment, cell adsorption, and genome ejection. Our structural analysis suggests that the baseplate can be subdivided into three subcomplexes encoded by Open Reading Frames (ORFs) 36–46, which likely assemble sequentially.

First, the baseplate cap (Fig. 5a and Supplementary Fig. 2) formed by gp36, gp37/gp38, gp41, and gp43 assembles onto the repetitive tail tube, sealing the genome-ejection channel. Gp36 (or tail tube-B, res. 1–167) is similar to the tail tube gp32 (RMSD 3.6 Å) and forms a hexameric ring concentric to gp32. This factor provides an attachment surface to three copies of the gp37/gp38 heterodimer (res. 1–179 and res. 1–197, respectively) that generate a threefold symmetric complex under the gp36 hexamer. This symmetry mismatch allows attachment of the trimeric gp41 (res. 1–287) that restricts the lumen providing an anchoring point to the trimeric tail tip gp43 (res. 1–221), which seals the tail. The E217 tail tip is like other *Myoviridae* tips and consists of a triple β-helix fold[46] 99% identical in amino acid sequence to the cell-puncturing protein ORF41 from *Pseudomonas* phage SN (PDB 4RU3).

The second set of proteins in the E217 baseplate includes the adaptor subunits gp33 (res. 1–107) and gp34 (res. 1–116) (Fig. 5b and Supplementary Fig. 2). The first subunit, gp33, binds the C-terminal ~40 amino acids of gp37 and gp38 that extend outward, while gp34 binds the β-sheet ridge formed by gp37/gp38, generating a hexameric ring above gp33. Notably, both gp33 and gp34 form hexameric assemblies that lack intra-subunit contacts suggesting these proteins are adaptors instead of discrete structural components of the baseplate.

The third subcomplex in the E217 baseplate, or nut (Fig. 5c), is built by six copies of the triplex complex (Supplementary Fig. 8a) that

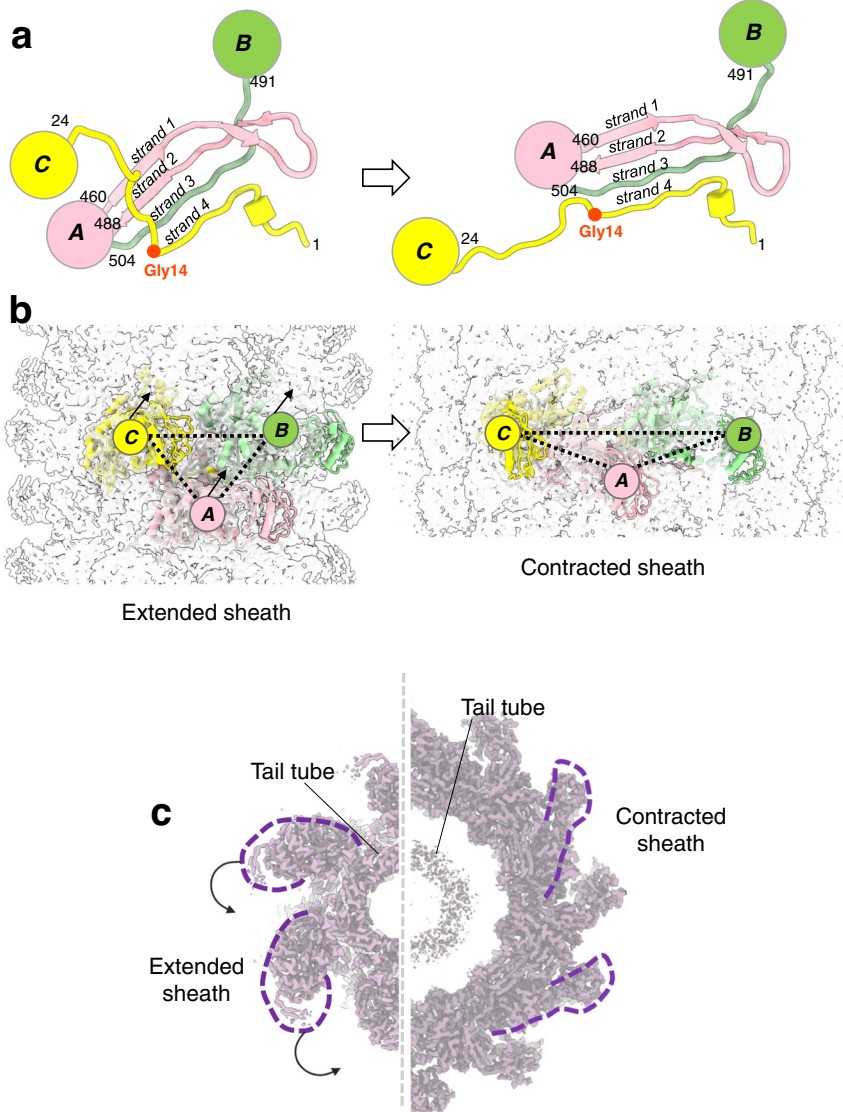

**Fig. 4 | Conformational changes in the sheath protein. a** Magnified view of the intermolecular β-sheet formed by three sheath proteins, referred to as subunits A, B, and C. On the left side is the association in the extended tail before contraction, while on the right panel is the same secondary structure element after tail contraction. In both panels, subunit A exposes the β-turn spanning residues 460–488

(e.g., β-strands #1 and #2, colored in pink) that hydrogen bond with subunit B C-terminal β-strand #3 (green) and subunit C N-terminal β-strand #4 (yellow). **b** Quaternary structure representation of the three sheath subunits A, B, and C described in (**a**). **c** A section view of the E217 sheath in the extended versus contracted states.

assemble at the bottom of the baseplate surrounding the cap complex. Each triplex complex consists of two conformers of gp44 (res. 1–417), referred to as gp44-a and gp44-b, and one gp45 (res. 1–504), slightly larger than gp44 (Supplementary Fig. 8a). Notably, gp44-a and gp44-b are the same protein but in drastically distinct conformations, with the conformer gp44-a being globular and gp44-b extended, like beads-on-a-string (Supplementary Fig. 8b). The interface area between gp45:gp44-a is significantly smaller than that of gp45:gp44-b (1712.5 Å² versus 2801.2 Å²), suggesting the two conformers exist as a result of gp44's intrinsic plasticity and differential binding contacts with gp45.

### E217 tail fibers attach flexibly to the baseplate nut

The baseplate nut provides an anchoring site for tail fibers (Fig. 5d), which had poor density in the cryo-EM reconstruction of the extended tail. However, a localized reconstruction of the E217 tail fiber revealed density for the N-terminal portion of the fiber. The estimated resolution of this density was limited to 7.3 Å (Fig. 5e), insufficient to build a de novo model. We then generated a prediction model of the trimeric gp46 using AlphaFold2 (Supplementary Fig. 9a)[47]. ORF46, located

immediately after the triplex subunit gp45, encodes a large protein of 946 amino acids, E217 tail fiber. We fit the predicted fiber into the density (Fig. 5e), with the N-terminal 340 amino acids contacting the triplex subunit gp45, which gave an excellent fit to the localized reconstruction (Fig. 5e) that we further improved by real-space refinement. At the same time, gp46 residues 341–964 had no visible density in the cryo-EM reconstruction due to their flexibility. This fiber region folds into five globular domains, highly enriched in glycine residues (Supplementary Fig. 9a) that likely move like an octopus tentacle.

The E217 tail fiber attaches to the triplex subunit gp45 (Supplementary Fig. 9b). The association between these two proteins is intimate. A long loop in gp45 spanning residues 76–123 (referred to as gp45 fiber attachment loop) (Fig. 5c) fits snugly at the trimeric helical interface generated by gp46 N-termini (Supplementary Fig. 9b), which adopts a coiled-coil structure similar to the tail needle gp26[48]. The total surface buried upon binding is 1125 Å² and is rich in hydrogen bonds and van der Waals contacts (Supplementary Table 1). At neutral pH, the tail fibers alternate a downward and upward orientation (Fig. 1a),

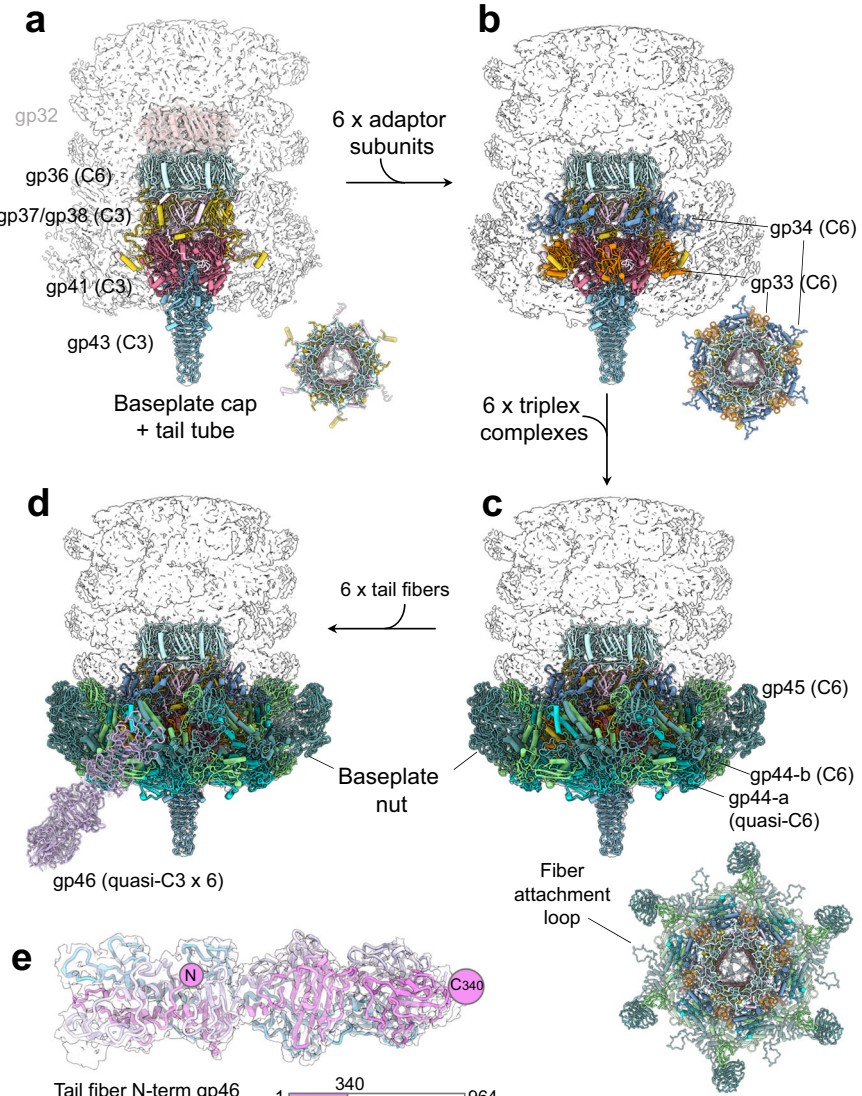

**Fig. 5 | Hierarchical organization of the E217 baseplate in the extended conformation. a** 3.4 Å cryo-EM map with overlaid all baseplate cap components, namely, gp36, gp37/gp38, gp41, and gp43. **b** Baseplate as in (**a**) with the addition of (**b**) adaptor subunits gp33 and gp34 and **c** baseplate nut formed by six triplex subunits. **d** Localized reconstruction of one tail fiber docked to the extended baseplate. **e** A magnified view of the trimeric tail fiber gp46 (res. 1–340) is overlaid with the threefold averaged density, calculated at 3.4 Å resolution at 0.143 FSC and displayed at 4 σ.

suggesting that the gp45 fiber attachment loop allows for gp46 rotation.

**Conformational changes in the baseplate upon genome ejection**
Alkaline pH triggers E217 tail contraction and baseplate dissociation into two components (Fig. 6a). The baseplate cap subunits gp36, gp37/gp38, and gp41 remain at the distal tail tip, losing both the adaptor factor gp33 and the tail tip gp43 (Fig. 6b). In contrast, the baseplate nut (e.g., subunits gp44-a, gp44-b, gp45, and gp46) moves upwards along the tail tube like a bolt around a nut, firmly attached to the last ring of sheath proteins (Fig. 6c). Notably, all six tail fibers point downward in the contracted state while alternate up- and downward positions in the extended tail (Fig. 1a, b).

Comparing the structures of triplex complexes in the baseplate nut before and after contraction provides clues to decipher the mechanisms of tail contraction. In the extended tail (Fig. 7a), gp44-a and gp44-b make a handshaking contact at the baseplate rim. The flexible pin domain of gp44-a (Supplementary Fig. 8b) is rotated about 70° from its C-terminal domain (Fig. 7b), and six pin domains extend outward into the channel to make contact with the tail tip. In contrast,

in the contracted state, gp44-a and gp44-b slide away from each other, making a finger-tip contact at the baseplate rim (Fig. 7c). In turn, the pin domain moves ~35° counterclockwise (Fig. 7d), folding onto its C-terminal domain, and losing bonding interaction with the tail tip. Rotation of pin domains away from the tail tube causes lumen expansion and tail tip release (Fig. 7e). As previously pointed out, the contracted sheath makes no contact with the tail tube (Fig. 4c), which does not change conformation upon contraction due to its rigidity.

**Identification of the E217 host receptor**
We isolated a spontaneous PAO1 mutant resistant to E217, which we named 4a in Fig. 8a. E217 was unable to adsorb to 4a (Fig. 8b), suggesting that the 4a mutant may have an altered or absent phage receptor. Since many *Pseudomonas* phages characterized so far exploit the host lipopolysaccharide (LPS) as receptor[49–51], we analyzed the 4a LPS by gel electrophoresis. We found that the bands corresponding to LPS species capped with the O-antigen, present in the LPS of the parental PAO1 strain (e.g., the Cm and Cl bands in Fig. 8c), were absent in the 4a LPS (Fig. 8c). Moreover, pre-incubation of E217 with the LPS extracted from PAO1, and not from 4a, impaired phage infection

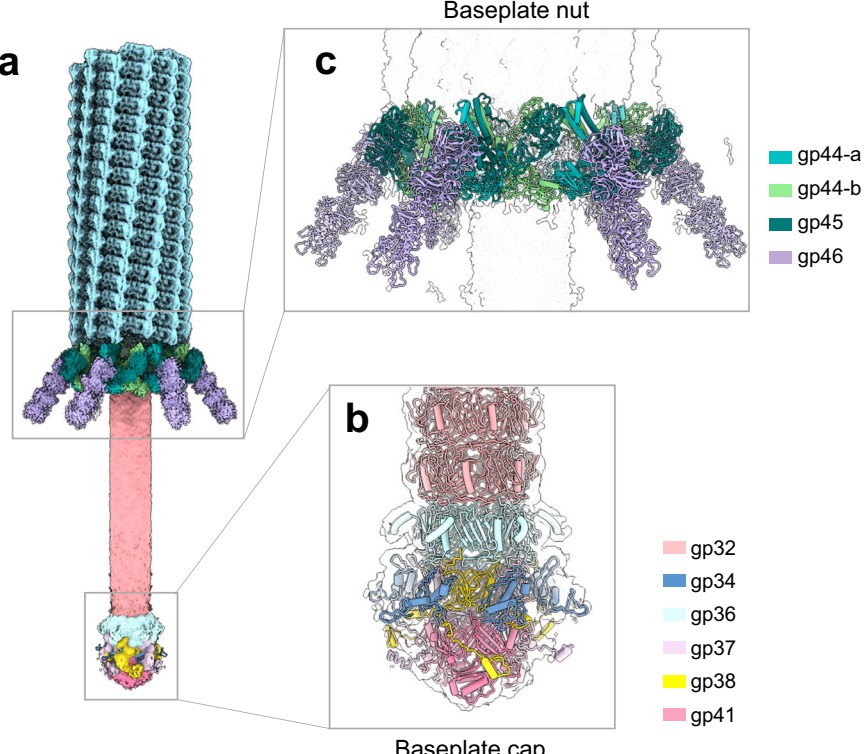

Baseplate nut

**c**

gp44-a
gp44-b
gp45
gp46

**a**

**b**

gp32
gp34
gp36
gp37
gp38
gp41

Baseplate cap

**Fig. 6 | Architecture of the contracted baseplate. a** The density map of the contracted baseplate calculated at 5.3 Å displayed at 2.5 σ. Its component proteins separate into two parts: the baseplate cap (devoid of the tail tip) remains at the bottom (**b**), while the baseplate nut (**c**) moves up with the contracted sheath. All tail fibers are in downward conformation after contraction.

(Fig. 8d), most probably because the phage receptor-binding proteins (RBPs) bound PAO1 LPS and were unable to interact with the 4a LPS. Overall, these data are consistent with the O-antigen moiety of the LPS as the E217 phage receptor.

Domain annotation analysis of E217 tail fiber identified an N-terminal carbohydrate-binding domain (CBD), a lectin-binding domain and a second CBD in the middle of the fiber, and a C-terminal α-amylase domain (Supplementary Fig. 9a). CBDs can bind LPS units, while the lectin-binding domain is a putative O-antigen binding receptor, as suggested for R-type pyocin tail fibers[52]. Lectin-binding domains contain a jellyroll fold with a variable distal loop network implicated in binding to LPS O-antigen. In contrast, the α-amylase domain belongs to the class of O-Glycosyl hydrolases, widespread enzymes that hydrolyze the glycosidic bond between two or more carbohydrates or between a carbohydrate and a non-carbohydrate moiety[53] commonly found in *Podoviridae* tailspikes and tail fibers from many lactophages[13]. However, E217 fiber lacks LPS hydrolytic activity in vitro.

## Discussion

Infections caused by the gram-negative pathogen *P.aeruginosa* are a leading cause of morbidity and mortality worldwide. Phage therapy against *P.aeruginosa* has gained attention as a promising therapeutic weapon in the fight against CF-related infections[54,55].

E217 is a PB1-like phage part of an experimental phage cocktail to eradicate *P. aeruginosa* infections in vivo[1–3]. In our study, we describe the atomic structure of the E217 phage, which we solved by combining the power of icosahedral and localized reconstructions.

We found that E217 has a $T = 9$ icosahedral capsid decorated by gp24, a likely cementing protein stabilizing the E217 capsid[56]. We defined the atomic structures of all neck components, namely the head-to-tail, collar, and gateway factors, which connect the icosahedral head to the contractile tail. By comparing extended versus contracted

tails, we determined that the neck does not undergo global conformational changes due to contraction, except for the portal barrel that disappears after genome ejection and the gateway C-terminal extension that rotates to allow for sheath contraction. We deciphered the tertiary and quaternary arrangement of the E217 sheath protein that forms a complex intermolecular mesh. The E217 sheath protein resembles topologically the T4 counterpart[43], which also consists of three domains like E217 gp31. However, it is vastly different in amino acid sequence and structure, underscoring how a similar tertiary structure can be conserved in different viruses with minimal sequence identity[57]. Sheath subunits make extensive lateral contacts in the contracted state, providing the energetic contribution that makes this conformation thermodynamically stable. In contrast, sheath subunits are porous in the extended conformation and held together only by an intermolecular β-sheet. Thus, E217 sheath proteins create a cross-linked two-dimensional mesh with similar connectivity in the extended and contracted state of the sheath[22,23] but with different topology and outside diameters.

We deciphered the complete architecture of the E217 baseplate that we rationalized as two complexes, an idle cap complex that seals the distal end of the tail tube and a mobile complex, the nut, which attaches to the cap via adaptor factors, gp33, and gp34. Interestingly, all four proteins in the cap complex, gp36, gp37, gp38, and gp41, are evolutionarily related to the tail tube and adopt a similar fold, consisting of two orthogonally packed β-sheets that form the inner lumen of the tail[46]. The symmetry fallout from the sixfold tail tube to the trimeric tail tip is transmitted through the heterodimeric gp37/gp38 complex, which also recruits the adaptors gp33 and gp34. The baseplate nut assembles outside the cap like a crown and remains bound to the sheath after contraction. The nut moves along the tail tube during contraction, losing the adaptor factor gp33, while concomitantly, the tail tip gp43 and viral genome are ejected. The plasticity in the triplex complex reflects primarily gp44-a beads-on-a-string conformation,

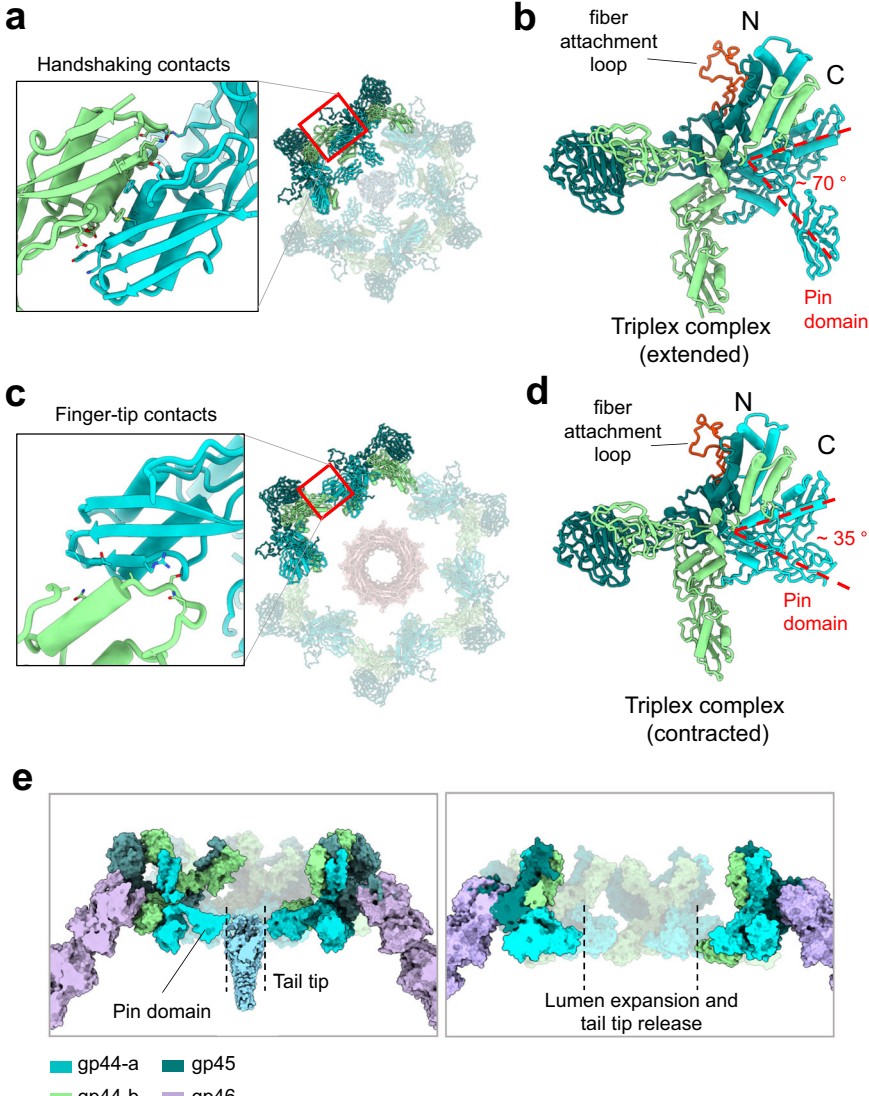

**Fig. 7 | Conformational plasticity of the triplex complex. a** In the extended conformation, neighboring triplex complexes make handshaking contacts with gp44-a pin domains projecting toward the tail tip. **b** Gp44-a pin domain is ~70° away from the C-terminal domain. **c** In the contracted conformation, neighboring triplex complexes make finger-tip contacts, and **d** gp44-a pin domains swing away from the tail tip, collapsing onto their C-terminal domains. **e** Section views of the base-plate before (left) and after (right) sheath contraction. Pin domain retraction causes lumen expansion and tail tip release.

which can adopt two *quasi*-equivalent conformations before and after contraction. This observation effectively extends Caspar and Klug's principle of *quasi*-equivalence[31], well established for capsid proteins, to a baseplate factor conserved in *Myoviridae*. Same as viral capsid proteins can adopt locally similar but non-identical positions across the capsid surface, allowing a single protein to close into an icosahedral cage, the *quasi*-equivalence of E217 triplex subunit gp44-a determines two ways to cement the tail tube.

The results presented in this paper allow us to formulate a model for how E217 attachment to the *Pseudomonas* surface triggers the ejection of the phage genome through the tail tube[14]. We envision four steps in this infection process (Fig. 9a).

Step 1: E217 tail fibers are in constant Brownian motion, waving up and down, as seen in the reconstruction of E217 with an extended tail. Unlike T4, E217 has only one type of tail fiber, built like beads on a string. Tail fibers function like a probing device, stochastically fluctuating to enhance the chance of encountering a receptor.

Step 2: The association of one gp46 fiber with the host O-antigen via a lectin-binding domain (Supplementary Fig. 9a) results in phage attachment to the host surface. This reduction of 3d dimensionality triggers a cooperative absorption cascade, whereby other fibers fluctuate downward and contact the host LPS, as proposed for T4 short fibers[19].

Step 3: After at least two-tail fibers have attached to the host surface, a mechanic signal is transmitted from gp46 to the triplex complex via the flexible fiber attachment loop in gp45 (Supplementary Fig. 9b). Specifically, lateral pulling of gp45 causes a change in the gp44-a/b association from handshaking to finger-tip contacts and collapsing of the gp44-a pin domain. In the extended metastable conformation, the gp44-a pin domain makes weak contact with the tail tip (Fig. 9b, left), retaining the tail tip in the cap complex and stabilizing the extended sheath.

Step 4: Following fiber attachment and gp45 lateral pulling, the gp44-a pin domain retracts away from the tail tip, collapsing against the gp44-a C-terminal domain and obliterating the only point of contact between the nut and tail tube (Fig. 9b, right). As a result, the nut, formed by six triplex complexes, becomes uncoupled from the tail tube and snaps upward like a nut around a bolt. The driving force of sheath contraction is the formation of intermolecular bonds among sheath proteins that release energy by aggregating in a 3D meshwork[26].

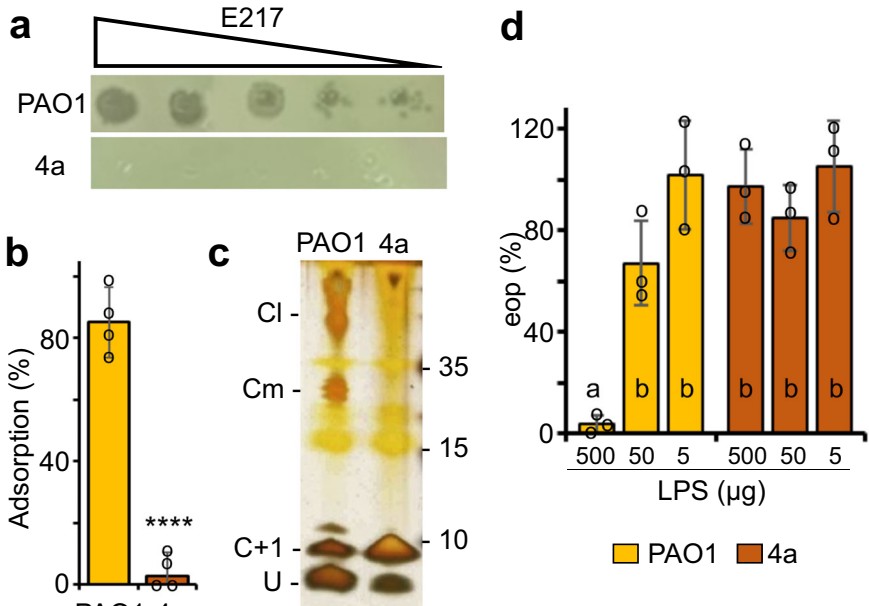

**Fig. 8 | E217 associates with *P. aeruginosa* O-antigen. a** E217 does not form plaques on the 4a mutant. Serial dilutions (×10) of an E217 stock were replicated onto PAO1 or 4a and used as indicators. **b** E217 (yellow bar) has defective adsorption on 4a (brown bar). Bars represent averages (*N* = 4) with standard deviation. ****P* < 0.0001 as estimated with a two-tail *T* test, with *P* = 7.26E-05. **c** The 4a mutant has defective LPS lacking the Cm and Cl forms, namely the LPS with medium and long chain O-antigen. U, uncapped LPS; C + 1, LPS capped with a single O-antigen repeat[63,79]. The position of MW markers is indicated on the left in kDa. **d** Pre-incubation of E217 with LPS extracted from PAO1 (yellow bars) but not from 4a (brown bars) impairs phage infection. Bars indicate average eop (*N* = 3) with standard deviation. Bars denoted by the same letters are not significantly different according to ANOVA and Tuckey post-hoc analysis (with *F* = 18.03179 and *P* = 3.29E-05). Source data are provided as a Source Data file.

While tail fibers are anchored to the host LPS, sheath contraction allows the phage to move closer to the host surface, penetrating the periplasm with its tail tube. The host surface is a stator, and because the baseplate nut complex is tethered to tail fibers, the sheath contraction with tail fibers bound to LPS results in the insertion of the tail tube through the OM into the periplasm like a syringe needle. Sheath compression is permitted by the way the gateway protein holds onto the first layer of sheath proteins, forming a sturdy yet flexible β-sheet that can stretch to accommodate sheath expansion.

It remains unclear if the tail tube penetrates the IM providing a channel for genome ejection into the host cytoplasm, or if a transient hole is opened upon ejection of the tail tip by fusion of the tail tube with the inner membrane[19]. The genomes of *Myoviridae* like E217 remain largely unexplored, and it is also possible these phages encode ejection proteins[58] similar to T7 gp16 that are expelled through the tail tube to form an IM channel for genome ejection[59].

In summary, we have deciphered the architecture and design principles of a prototypical PB1-like phage used in an experimental phage therapy cocktail. We propose that the structural principles elucidated in this work are conserved in other *Myoviridae* of the widespread PB1 family. E217 complete structure and conformational dynamics shed light on *Pseudomonas* Myo-phages used for phage therapy, deepening the understanding of phages as biomedicines. We anticipate that the 3D-atlas of E217 structural proteins described in this paper will allow the mapping of resistance mutations and facilitate the identification of ORFs in related *Pseudomonas* phages used in phage therapy cocktails.

## Methods
### E217 phage preparation
*P. aeruginosa* strain PAO1[60] was grown at 37 °C to OD$_{600}$ = 0.5 (about $2.5 \times 10^7$ cfu ml$^{-1}$) in Luria Broth (LB) and infected with E217 phage[1] GenBank NC_042079) at a multiplicity of infection (MOI) of 0.001. Growth continued until cell lysis was detected as a drop in the OD$_{600}$.

The lysate was incubated 30 min at 37 °C with 1 µg ml$^{-1}$ of DNase and RNase before centrifugation at 5000×*g* for 10 min. After supernatant filtration through a 0.45 µm filter, 58 g L$^{-1}$ NaCl and 105 g L$^{-1}$ polyethylene glycol (PEG) MW 6 K were dissolved in the supernatant. The solution was stirred ~16 h at 4 °C before pelleting the phage particles at 20,000 × *g* at 4 °C 30 min. The pellets were resuspended in TN buffer (10 mM Tris-HCl, 150 mM NaCl pH 8), and the phage mixture was centrifuged on a cesium chloride step gradient from 1.3 to 1.6 g cc$^{-1}$ cesium chloride, top to bottom, formed in polyallomer ultracentrifuge tubes for Beckman rotor SW41. The phage mixture (3 ml) was applied to the top of the gradient, and gradients were centrifuged at 100,000 × *g* for 120 min at 4 °C in a Beckmann Optima XE-90 ultracentrifuge using an SW41 rotor. The phage bands, which usually sediment at the interface between the 1.5 g cc$^{-1}$ and 1/.6 g cc$^{-1}$ steps, were extracted from the tubes with a syringe, transferred into polyallomer tubes for SW60 Beckman rotor, and centrifuged ca. 16 h at 150,000 × *g* in a Beckmann Optima XE-90 ultracentrifuge using an SW60 rotor. The phage bands were collected as above and dialyzed 2× for 20 min against water and then 16 h against TN buffer, filtered through 0.22-µm filters, and stored at 4 °C.

### Adsorption assay
PAO1 strain and its 4a derivative, namely a spontaneous E217-resistant mutant, were used. In total, 03 × 10$^9$ cfu of PAO1 or 4a were incubated 10 min at 37 °C with ca. 08 × 10$^3$ plaque-forming unit (pfu) of E217 in 1 ml of LB. The bacterial cells with adsorbed phages were pelleted by centrifugation at 5000×*g* for 10 min, and the free phage in the supernatant was titred using PAO1 as an indicator. The adsorption efficiency (%) was calculated according to the formula [1-(free phage/phage input)] × 100.

### LPS extraction
PAO1 and 4a cultures were grown in 50 ml of LB at 37 °C to OD$_{600}$ = 0.8 and centrifuged 15 min at 5000×*g*. The LPS was extracted from

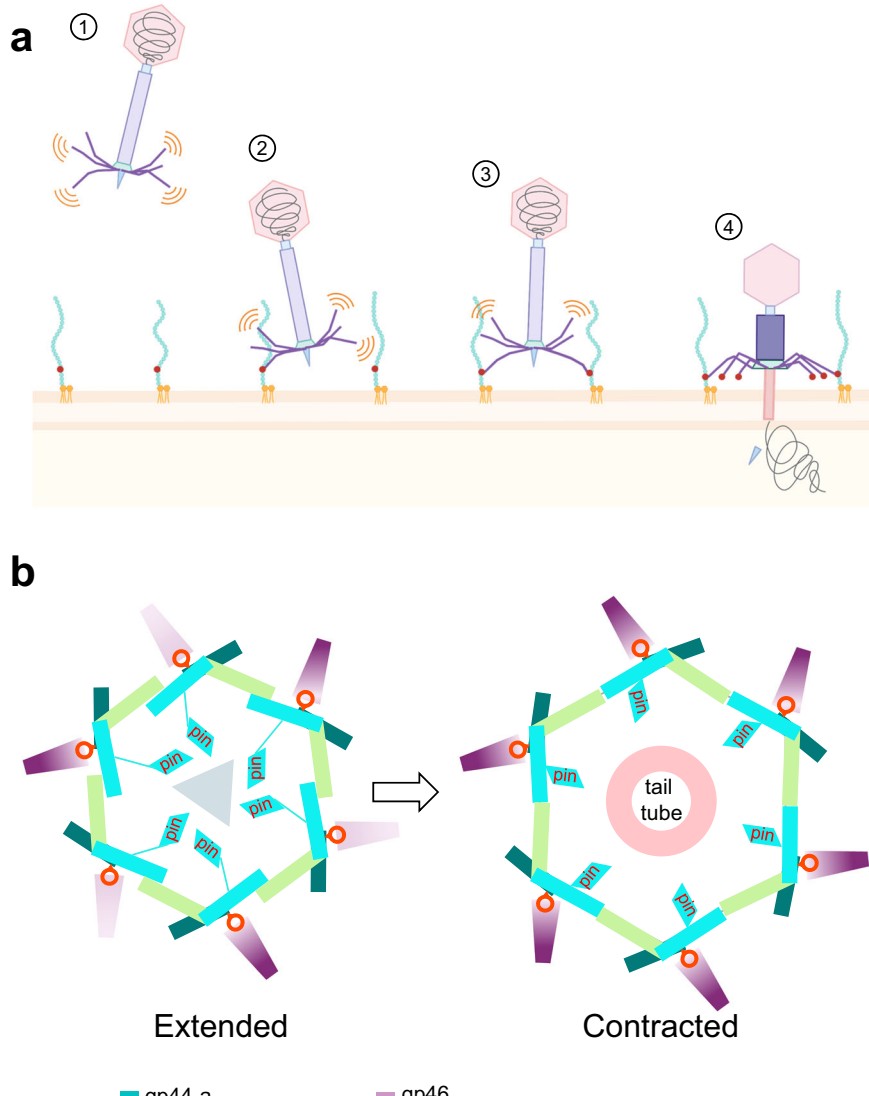

**Fig. 9 | Model for signal transduction. a** Schematic representation of E217 infecting *P. aeruginosa*. Four proposed steps of infection are shown. Each step is accompanied by distinct conformations of the tail fibers and conformational changes in the baseplate, shown in the bottom panels. **b** Diagram of the E217 baseplate (viewed from the bottom) in the extended (left) and contracted (right) conformations, emphasizing the position of gp44-a pin domain. Tail fibers are colored in purple: light and dark purple represent tail fibers projecting below and above the plane. The images were created with BioRender.com.

bacterial pellets with hot phenol and diethyl ether as described in ref. 61. After the extraction, LPS samples were lyophilized for ~20 h, and the pellets were weighed and resuspended in TN buffer at 10 mg ml⁻¹. The LPS (5 μg) was visualized by 18% tricine SDS-PAGE followed by silver staining[62,63].

### LPS binding assay
To evaluate LPS binding, $10^3$ pfu of E217 were incubated with 500 μg, 50 μg, 5 μg of LPS extracted from PAO1 or 4a or mock-incubated without LPS for 60 min at 37 °C in a final volume of 0.15 ml of TN buffer. The samples were mixed with 0.3 ml of overnight PAO1 culture and 2.5 ml of soft agar and plated onto an LB agar plate. The efficiency of plating (eop) was measured by counting the plaques after overnight incubation at 37 °C and normalizing it to the number of plaques obtained in mock-incubated samples.

### Vitrification and data collection
In all, 2.5 μl of E217 mature virions at $1 \times 10^{14}$ phages ml⁻¹ was applied to a 200-mesh copper Quantifoil R 2/1 holey carbon grid (EMS) previously glow-discharged for 60 s at 15 mA using an easiGlow (PELCO). The grid was blotted for 7.5 s at blot force 2 and vitrified immediately in liquid ethane using a Vitrobot Mark IV (FEI). Cryo-grids were screened on a Thermo Scientific Glacios cryo-transmission electron microscope equipped with a Falcon 4 detector and operated with TEM interface (Talos v7.x, TEM v7.X, TIA v5.X, FluCam Viewer v7.X), at Thomas Jefferson University. EPU (v2.X) software was used for data collection using accurate positioning mode. For high-resolution data collection, micrographs were collected on a Titan Krios microscope operated at 300 kV and equipped with a K3 direct electron detector camera (Gatan) at the National Cryo-Electron Microscopy Facility (NCEF) at the Frederick National Laboratory, MD. Micrographs were collected in counting mode with an energy filter of an image pixel size of 1.12 Å, a nominal ×81,000

magnification, a total dose of 50 e/Å², 40 frames, and defocus range of −0.75 to −1.5 μm. To trigger sheath contraction, phage E217 at $1 \times 10^{14}$ phages ml⁻¹ was incubated overnight in 0.1 M Carbonate-Bicarbonate buffer (pH = 11.4) at 4 °C. The contracted E217 was then dialyzed against PBS buffer (pH = 7.5) of a ratio of 1:1000 overnight at 4 °C and concentrated in a spin concentration unit of 100 kDa cut-off. The vitrification procedures of contracted E217 were the same as the E217 at neutral pH described above. High-resolution data for contracted E217 were collected at the National Center for Cryo-EM Access and Training (NCCAT). Additional data collection parameters are in Table 1.

## Cryo-EM single particle analysis of E217 virions with extended and contracted tail

Multi-frame movies of native phage E217 were motion-corrected with MotionCor2[64], yielding 22,015 micrographs. RELION's implementation of motion correction was applied to the micrographs with options of dose-weighted averaged micrographs and the sum of non-dose-weighted power spectra every 4 e⁻/Å². CTF (Contrast Transfer Function) estimation was carried out using CTFFIND4[65]. After an initial round of reference picking and 2D classification, 15,517 virion particles were subjected to a reference-free low-resolution reconstruction without imposing symmetry. The particles were then 3D classified into four classes with I4 symmetry imposed. Of the four classes, one class of 15,505 particles had the most resolved reconstruction and was subjected to 3D auto-refined to align the particles finely. The particles were then expanded according to I4 symmetry using RELION's relion_particle_symmetry_expand function to relax particle orientations to be used in the following localized reconstruction steps. A cylindrical mask (radius = 150 Å) was generated using SCIPION 3.0[66] and then resampled onto a reference map covering the fivefold vertex in UCSF Chimera[67]. The cylindrical mask was then used for non-sampling 3D classification without imposing symmetry to search for the tail. Locally aligned particles were then combined from two classes, and duplicates were removed, yielding 13,257 particles. The initial localized reference map was reconstructed directly from one of the classes using RELION's relion_reconstruct function. Tail-aligned particles were further auto-refined, imposing C5 and C6 symmetry, aimed at aligning the capsid and the long tail, respectively. The C5/C6 aligned particles were subjected to a third 3D classification, removing heterogeneous particles to 10,826. Final aligned localized tail particles were CTF refined and polished. Symmetries of tail-tube components presented in this study were derived by inspection of the electron density maps and by association with prior studies on other *Myoviridae* tail tubes[22] rather than ab initio from the analysis of power spectra of the E217 tail tubes.

## Cryo-EM single particle analysis of E217 baseplate

To reconstruct the baseplate, we manually picked 12,442 particles containing the distal tip of the E217 tail and reconstructed the baseplate with C6 symmetry. A C6 symmetry expansion and a search for C3 features for the baseplate were then carried out. All steps of SPA, including 2D/3D classification, 3D refinement, CTF refinement, particle polishing, post-processing, and local-resolution calculation, were carried out using RELION 3[68,69]. The final densities were sharpened using phenix.auto_sharpen[70]. RELION_postprocess[68,69] was used for local-resolution estimation, and drawings of electron density maps and local-resolution maps were generated using ChimeraX[71]. Contracted sheath. To obtain a high-resolution reconstruction of the E217 contracted sheath, empty particles with contracted tails were picked from 22,714 movies collected on a 300 kV Krios/K3 and processed as detailed above. After extensive particle picking and 2D classification, 10,126 contracted tail particles were reconstructed in C6 symmetry.

## De novo model building and refinement

All de novo atomic models presented in this paper, except the tail fiber N-terminal fragment, were built de novo using Coot[72] or Chimera[67] and refined using several rounds of rigid-body, real-space, and B-factor refinement using phenix.real_space_refinement[73]. All final models were validated using MolProbity[74] (Table 1). We used six different maps for model building (Supplementary Fig. 1c): (i) a 2.8 Å map combined of icosahedral and localized reconstruction of the mature head that revealed the atomic structure of the E217 capsid- and decorating protein; (ii) a 3.3 Å C12-averaged localized reconstruction used to build the dodecameric portal protein bound to twelve copies of the head-to-tail adaptor; (iii) a 3.4 Å C6-averaged localized reconstruction used to build hexameric models of the collar, gateway, and tail tube; (iv) 3.1 Å C6 densities of extended or contracted sheaths used to model sheath proteins; (v) a 3.4 Å C6/C3 of the extended baseplate that allowed modeling all 66 polypeptide chains; (vi) a 4.5 Å C6/C3 of the contracted baseplate used to place different baseplate components manually, and then refined using the fit-into-map command in Chimera[67] and phenix.real_space_refinement[73]. Finally, the tail fiber N-termini (res. 1–340) was modeled in a focused asymmetric reconstruction generated from the C6 symmetry expanded map of the baseplate, using a cylindrical mask covering only one tail fiber. The full-length tail fiber was modeled in fragments using AlphaFold2[47]. The first 340 amino acids were manually placed in the 3.6 Å focused reconstruction of the tail fiber, and rigid- and real-space refined, yielding a final CC = 0.84 (Table 1).

## Structure analysis

All ribbon and surface representations were generated using ChimeraX[71] and PyMol[75]. Structural neighbors were identified using the DALI server[32]. Binding interfaces were analyzed using PISA[76] and PDBsum[77] to determine bonding interactions and interatomic distances. Supplementary Table 1 provides a comprehensive list of interactions at symmetry-mismatched interfaces. Domain annotation for gp46 was done using SMART[78]. RMSD between superimposed PDBs was calculated using SuperPose Version 1.0 (superpose.wishartlab.com) and Matchmaker in ChimeraX[71]. The Coulombic Electrostatic Potential was calculated and displayed with surface coloring using ChimeraX[71].

## Reporting summary

Further information on research design is available in the Nature Portfolio Reporting Summary linked to this article.

## Data availability

Atomic coordinates for E217 capsid:decorating protein complex, portal:head-to-tail:collar:gateway complex, extended sheath:tube complex, contracted sheath, extended baseplate complex have been deposited in the Protein Data Bank under accession codes 8FRS, 8FVH, 8FUV, 8FVG, and 8EON, respectively. The cryo-EM density maps generated in this study have been deposited in Electron Microscopy Data Bank database under accession code EMD-29406, EMD-29487, EMD-29481, EMD-29486, and EMD-28405. Source data are provided with this paper.

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

## Acknowledgements

We thank the staff at NCEF and NCCAT for their assistance in cryo-EM data collection. We thank Drake Schaefer at Thomas Jefferson University for his support in 3D printing. This work was supported by the National Institutes of Health grants R01 GM100888, R35 GM140733, and S10 OD030457 to G.C., and by the Fondazione per la ricerca sulla Fibrosi cistica-Associazione Trentina Fibrosi Cistica ODV In ricordo di Pio Nicolini grant FFC#15/2021 to F.B. Research in this publication includes work carried out at the Sidney Kimmel Cancer Center X-ray Crystallography and Molecular Interaction Facility at Thomas Jefferson University, which is supported in part by National Cancer Institute Cancer Center Support Grant P30 CA56036. A portion of this work was carried out at NCEF (supported by contract 75N91019D00024); NCCAT and the Simons Electron Microscopy Center, supported by grants from NIH (U24 GM129539), Simons Foundation (SF349247) and NY State Assembly.

## Author contributions

F.L., C.-F.D.H., R.K.L., R.Y. and G.C. performed all steps of cryo-EM and structural analysis, deposition of atomic coordinates and maps, and figure preparation. F.F. and F.B. purified the E217 virion and completed the identification of the host receptor assay. G.C. and F.B. supervised the project and wrote the paper. All authors contributed to the writing and editing of the manuscript.

## Competing interests

The authors declare no competing interests.
