## [Peer Review File · Nature Communications]

High-resolution cryo-EM structure of the Pseudomonas bacteriophage E217REVIEWER COMMENTS

Reviewer #1 (Remarks to the Author):

Li et al. describes the structure and assembly of the tail tube of the *Pseudomonas* bacteriophage E217. The authors dissect the mechanism of the tail contraction for genome release into the host by using high-resolution cryo-EM. In doing so they determine the three-dimensional structure of 19 gene products. Bacteriophage E217 belongs to the Myoviridae family and most of the structural information of tail tubes of Myoviridae is derived from the wealth of data gathered on bacteriophage T4. Nevertheless, the differences in genome size and the apparent dissimilarity at genome level between the two phages prompted this study. The paper also provides information on the identification of the host receptor.

The manuscript is elegantly presented and the breadth of the solid findings on the tail-contraction and genome release mechanism sufficiently unique to warrant its publication in *Nature Communications*.

There are however minor issues that require some clarification:

(1) page 2, line 111: Instead of 'This paper...' could the Author use 'Our study....' To minimize confusion with the previous lines.

(2) page 3, lines 153-154: the RMSD 8.8 Å – the authors report that DALI identifies the MCP of Siphovirus TW1 (PDB 5WK1) the 'most similar' structure to gp25 with an RMSD of 8.8 Å. Would it be possible to have a figure of the superimposition as a supplementary panel? Furthermore, it would be beneficial to provide not only the RMSD but also the number of Calphas that have superimposed out of the total number of Calphas that each structure is composed of.

Related to this issue – if one looks at Supplementary Figure 3b, the RMSD is 5.7 Å, but what would the RMSD be if the E-loop and the first 15 N-terminal aa are removed? The point here is to distinguish the possible differences within the core structure across the independent gp25 forming the IAU, and the more flexible extensions that likely change according to the local environment and are responsible for the relatively high (in the opinion of this reviewer) 5.7 Å RMSD.

The issue of the RMSD values with the addition of Calphas superimposed should be revisited throughout the text and in the Supplementary Figure 6. Indeed, the values of RMSD provided for the E217 sheath protein versus corresponding Pyocin and T4 are quite high - 21 and 26 Å, respectively. And yet in the case of Pyocin if one removes from the domain A of E217 gp31 and re-runs the superimposition what would the new RMSD be? And how many Calphas would superimpose?

(3) The tail-tube structure machinery is examined with precision and clarity. Having said that, it would be beneficial to add a couple of lines in the 'Materials and Methods' section regarding the prior knowledge available for the application of the different symmetries to the distinct structures of the tail tube. The 'Introduction' provides references to build a background knowledge on the targeted system, but it would be useful to make explicit that symmetries were derived by association with prior knowledge on Myoviridae rather than 'ab initio' from the analysis of power spectra of the E217 tail tubes.

(4) regarding the symmetry – I would suggest adding the symbol of the symmetry also in the figures when possible. For example, in 'panel a' of Figure 2 a 'C12' in correspondence of gp19 and gp27 and a 'C6' on correspondence of gp28 and gp29...this would facilitate the grasping of the concept and role of the symmetry reduction proteins.

(5) page 6 lines 246 and 258: the use of 'dramatic' in close vicinity; would it be possible to use a synonym for one of them?

(6) page 7 line 307: '99% identical' – is this referring solely to the fold? Or also to the sequence?

(7) page 8 lines 320-322: is the 'cause' that sets off the folding into different structures/conformers, gp44-a versus gp44-b known? (steric hindrance or else??)

(8) page 10 line 414-416: can this sentence be clarified? Does 'fold' here refer to domain organization?

(9) page 11 lines 446-449: The step-1 is common to many phages and not necessarily specific to E217. Also, the whole tail tube contraction pathway summarized in Figure 9a is useful but it would be of more use if the differences with the T4 could be also schematically highlighted.

(10) page 12, lines 485-487: in the conclusion and throughout the text the Authors mention that this phage is used in an experimental phage cocktail to eradicate *Pseudomonas aeruginosa*. This biomedical application indeed boosts the relevance of the system in study. Beyond the general consensus that 'deepening the understanding of phage biomedicines is important to increase the safety (...)', could the authors briefly provide examples that support this conclusive sentence?

(11) page 22 – Table 1. Can the Authors also add in the table the number of frames (40) composing each movie?

Figures: well organized and clear but – something that nowadays is not often seen and yet still very useful for showing the quality of a map is a stereoview of the electron density of a structural details of one to the tail components. A further panel could be easily added in a supplementary figure.

PDB reports: OK

Thanks for the chance to review this interesting manuscript that reads amenably. The results have been carefully presented, interpreted, and discussed.

Reviewer #2 (Remarks to the Author):

The paper represents a tour-de-force structural investigation of a bacteriophage representing a genus that is important for human health, in particular for potential phage therapy applications. The presented cryo-EM data not only reveal structures of numerous individual proteins of the E217 phage but also structures of these proteins in both contracted and extended states of the virus, illuminating architectural and conformational reorganisations that occur during host recognition and infection.

While paper represents significant structural advance into our understanding of this clinically important group of bacteriophage, I feel it could be improved by taking the following points into account:

1) While Abstract suggests that paper reports 3.1-3.4 Å structures for both contracted and extended states (e.g. "Here, we describe the structure of the 42 whole E217 virion before and after DNA ejection at a resolution between 3.1-3.4 Å, 43 determined using cryogenic electron microscopy (cryo-EM)."), it appears that the resolution of the contracted state is only 4.5 Å (lines 137-138 "Treatment with alkaline pH triggered in vitro sheath 137 contraction 18, exposing ~500 Å of the tail tube (Fig. 1b), although the resolution of the 138 contracted virion devoid of DNA was limited to 4.5 Å."). This needs to be clarified in the Abstract to avoid confusion.

- 2) The Results section states "We solved the E217 head using icosahedral symmetry to 2.8 128 Å resolution." However, no validation report is provided for the icosahedrally averaged structure of the capsid, and the stats in Table 1 appear to be only for the decorating protein and not the major capsid protein? Furthermore, two of the five validation reports are labelled "Not for manuscript review".
- 3) Little (or none) information is provided about interaction of proteins (or their assemblies) at symmetry-mismatching interfaces.
- 4) The paper is rather thin in comparing structural findings with information available for other phage, in particular those belonging to Myoviridae (e.g. doi: 10.1038/emboj.2009.36; doi/10.1073/pnas.1605883113). Discussing structural data in a wider context is important for underlying significance of presented data.
- 5) Most importantly, it would be useful to discuss the proposed mechanism "A pathway for signal transduction from tail fiber to sheath contraction" in comparison with mechanisms proposed earlier on the basis of structural data for other representatives of Myoviridae.

Minor point:

I would abbreviate the Ramachandran plot statistics in Table 1 to the first decimal point (e.g. "7.1" instead of "7.11" etc).

Reviewer #3 (Remarks to the Author):

This is a quality work showing the high resolution structures of E217 pseudomonas phage used in phage therapy. Structures were solved before and after tail contraction to release the genome at resolutions allowing 19 new proteins to be built into phage neck, sheath, and baseplate as well as tail fibers. The structures are impressive.

At the end of the first paragraph of Results, ..."obtained by combining different cryo EM maps." What maps were combined to obtain what structures?

By definition, virion is an infectious virus particle that contains a genome. Thus, the contracted form of E217 is not a virion.

Supplemental Figure 2 may be important enough to bring into the paper as a Figure. The Atlas should include the pdb accession code with each protein. Figure 8 and 9 could move to supplemental.

All figures need more descriptive legends. The color coded key for Figures 2 and 6 is excellent, could it be used in other figures?

The first two paragraphs of Discussion summarize the results without any real discussion.

Point-by-point response to reviewer comments

We thank the reviewers and the editor for the thoughtful review of our paper. Below we provide a point-by-point response to all the reviewers' criticisms. All edits are marked in red in the revised manuscript.

Reviewer #1

Li et al. describes the structure and assembly of the tail tube of the Pseudomonas bacteriophage E217. The authors dissect the mechanism of the tail contraction for genome release into the host by using high-resolution cryo-EM. In doing so they determine the three-dimensional structure of 19 gene products. Bacteriophage E217 belongs to the Myoviridae family and most of the structural information of tail tubes of Myoviridae is derived from the wealth of data gathered on bacteriophage T4. Nevertheless, the differences in genome size and the apparent dissimilarity at genome level between the two phages prompted this study. The paper also provides information on the identification of the host receptor. The manuscript is elegantly presented and the breadth of the solid findings on the tail-contraction and genome release mechanism sufficiently unique to warrant its publication in Nature Communications.

Thank you!

There are however minor issues that require some clarification:

(1) page 2, line 111: Instead of 'This paper...' could the Author use 'Our study....' To minimize confusion with the previous lines.

Done! See page 2, line 114.

(2) page 3, lines 153-154: the RMSD 8.8 Å – the authors report that DALI identifies the MCP of Siphovirus TW1 (PDB 5WK1) the 'most similar' structure to gp25 with an RMSD of 8.8 Å. Would it be possible to have a figure of the superimposition as a supplementary panel? Furthermore, it would be beneficial to provide not only the RMSD but also the number of Calphas that have superimposed out of the total number of Calphas that each structure is composed of.

Done! On page 4, lines 161-162, we state: "**The RMSD between E217 and TW1 capsid proteins is 8.8 Å for 274 aligned Ca atoms out of 296 and 317 Cas, respectively (Supplementary Fig. 3d)**". A superimposition of TW1 and gp25 capsid proteins was added to **Supplementary Fig. 3d**.

Related to this issue – if one looks at Supplementary Figure 3b, the RMSD is 5.7 Å, but what would the RMSD be if the E-loop and the first 15 N-terminal aa are removed? The point here is to distinguish the possible differences within the core structure across the independent gp25 forming the IAU, and the more flexible extensions that likely change according to the local environment and are responsible for the relatively high (in the opinion of this reviewer) 5.7 Å RMSD.

Done! The RMSD for the capsid protein core (e.g., after deleting the E-loop and N-terminus) is 2.3 Å. We added this information to the **Supplementary Fig. 3b** legend.

The issue of the RMSD values with the addition of Calphas superimposed should be revisited throughout the text and in the Supplementary Figure 6.

Done! See page 6, lines 250.

Indeed, the values of RMSD provided for the E217 sheath protein versus corresponding Pyocin and T4 are quite high - 21 and 26 Å, respectively. And yet in the case of Pyocin if one removes from the domain A of E217 gp31 and re-runs the superimposition what would the new RMSD be? And how many Calphas would superimpose?

Done! See updated **Supplementary Fig. 6b**: "**The RMSD between E217 and pyocin sheath proteins is 21 Å for 39 aligned Ca atoms out of 504 and 385 Cas, respectively. Instead, the RMSD between E217 and T4 sheath proteins is 26 Å with only 5 aligned Ca atoms out of 504 and 479 Cas, respectively. After removing domain A from the E217 sheath protein, the RMSD between E217 and pyocin sheaths drops to 16 Å with 295 aligned Ca atoms but is still 26 Å for E217 and T4 sheaths (for 274 aligned Ca atoms)**".

(3) The tail-tube structure machinery is examined with precision and clarity. Having said that, it would be beneficial to add a couple of lines in the 'Materials and Methods' section regarding the prior knowledge available for the application of the different symmetries to the distinct structures of the tail tube. The 'Introduction' provides references to build a background knowledge on the targeted system, but it would be useful to make explicit that

symmetries were derived by association with prior knowledge on Myoviridae rather than 'ab initio' from the analysis of power spectra of the E217 tail tubes.

Done! We added a sentence on page 14, lines 592-595 "***Symmetries of tail-tube components presented in this study were derived by inspection of the electron density maps and by association with prior studies on other Myoviridae tail tubes (Ge et al., Nature, 2020) rather than ab initio from the analysis of power spectra of the E217 tail tubes.***"

(4) regarding the symmetry – I would suggest adding the symbol of the symmetry also in the figures when possible. For example, in 'panel a' of Figure 2 a 'C12' in correspondence of gp19 and gp27 and a 'C6' on correspondence of gp28 and gp29...this would facilitate the grasping of the concept and role of the symmetry reduction proteins.

Done! Symmetry symbols were added next to each protein in **Figures 2 and 5**.

(5) page 6 lines 246 and 258: the use of 'dramatic' in close vicinity; would it be possible to use a synonym for one of them?

Done! 'Dramatic' was replaced by "***significant***". See page 6, line 253.

(6) page 7 line 307: '99% identical' – is this referring solely to the fold? Or also to the sequence?

Yes, 99% refers to the sequence; consequently, the fold is virtually identical. On page 7, lines 316, we added, "***99% identical in amino acid sequence***".

(7) page 8 lines 320-322: is the 'cause' that sets off the folding into different structures/conformers, gp44-a versus gp44-b known? (steric hindrance or else??)

Done! Excellent point!! On page 8, lines 331-334, we added, "***The interface area between gp45:gp44-a is significantly smaller than that of gp45:gp44-b (1712.5 Å² versus 2801.2 Å²), suggesting the two conformers exist as a result of gp44 intrinsic plasticity and differential binding contacts with gp45***".

(8) page 10 line 414-416: can this sentence be clarified? Does 'fold' here refer to domain organization?

Done! We replaced 'fold' with '***tertiary structure***'. See page 10, line 425.

(9) page 11 lines 446-449: The step-1 is common to many phages and not necessarily specific to E217. Also, the whole tail tube contraction pathway summarized in Figure 9a is useful but it would be of more use if the differences with the T4 could be also schematically highlighted.

Done! We add a sentence on page 11, line 457: "***Unlike T4, E217 has only one type of tail fiber, built like beads on a string***". The reviewer is correct that describing the differences between E217 and T4 would add to this paper, but, unfortunately, the paper is already ~1,200 words over the recommended length for a full article in Nat Comms, and we cannot add more text without removing essential descriptions.

(10) page 12, lines 485-487: in the conclusion and throughout the text the Authors mention that this phage is used in an experimental phage cocktail to eradicate *Pseudomonas aeruginosa*. This biomedical application indeed boosts the relevance of the system in study. Beyond the general consensus that 'deepening the understanding of phage biomedicines is important to increase the safety (...)', could the authors briefly provide examples that support this conclusive sentence?

Done! The reviewer is correct. 'Safety' is the wrong word. We explained the importance of this study by stating (on page 12, lines 494-499) "***E217 complete structure and conformational dynamics shed light on Pseudomonas Myo-phages used for phage therapy, deepening the understanding of phages as biomedicines. We anticipate that the 3D-atlas of E217 structural proteins described in this paper will allow the mapping of resistance mutations and facilitate the identification of ORFs in related Pseudomonas phages used in phage therapy cocktails***".

(11) page 22 – Table 1. Can the Authors also add in the table the number of frames (40) composing each movie?

Done! We added this information to **Table 1**. We also revised the table to include data collection statistics for the E217 contracted tail. This information was previously only in the methods.

Figures: well organized and clear but – something that nowadays is not often seen and yet still very useful for showing the quality of a map is a stereoview of the electron density of a structural details of one to the tail components. A further panel could be easily added in a supplementary figure.

Done! We added new electron density information for each tail factor in **Supplementary Figs. 5a, 7a, and 8b**, which illustrate the quality of our cryo-EM maps. We did not use a stereoview representation to avoid overcrowding the (already crowded) supplementary figures.

PDB reports: OK

Thanks!

Thanks for the chance to review this interesting manuscript that reads amenably. The results have been carefully presented, interpreted, and discussed.

Thanks!

Reviewer #2

The paper represents a tour-de-force structural investigation of a bacteriophage representing a genus that is important for human health, in particular for potential phage therapy applications. The presented cryo-EM data not only reveal structures of numerous individual proteins of the E217 phage but also structures of these proteins in both contracted and extended states of the virus, illuminating architectural and conformational reorganisations that occur during host recognition and infection. While paper represents significant structural advance into our understanding of this clinically important group of bacteriophage, I feel it could be improved by taking the following points into account:

Thank you!

While Abstract suggests that paper reports 3.1-3.4 Å structures for both contracted and extended states (e.g. “Here, we describe the structure of the 42 whole E217 virion before and after DNA ejection at a resolution between 3.1-3.4 Å, 43 determined using cryogenic electron microscopy (cryo-EM).”), it appears that the resolution of the contracted state is only 4.5 Å (lines 137-138 “Treatment with alkaline pH triggered in vitro sheath 137 contraction 18, exposing ~500 Å of the tail tube (Fig. 1b), although the resolution of the 138 contracted virion devoid of DNA was limited to 4.5 Å.”). This needs to be clarified in the Abstract to avoid confusion.

Done! Revised on page 1, line 43-44.

The Results section states “We solved the E217 head using icosahedral symmetry to 2.8 128 Å resolution.” However, no validation report is provided for the icosahedrally averaged structure of the capsid, and the stats in Table 1 appear to be only for the decorating protein and not the major capsid protein?

Done! We used icosahedral symmetry only as an initial reconstruction step to calculate a C5 localized reconstruction of the capsid. The actual coat and cementing protein models were built de novo in a C5 localized reconstruction. Therefore, we revised the sentence: “**We reconstructed the E217 head using a localized reconstruction with five-fold symmetry imposed (C5), which yielded 2.8 Å resolution at $FSC_{0.143}$.**” See page 3, lines 131-133.

Furthermore, two of the five validation reports are labelled “Not for manuscript review”.

Validation reports of 8EON and 8FRS were re-uploaded. All five reports are now ‘*For manuscript review*’.

3) Little (or none) information is provided about interaction of proteins (or their assemblies) at symmetry-mismatching interfaces.

Done! We have added **Supplementary Table 1** to illustrate the symmetry-mismatching interfaces of Head-to-Tail:collar, Tail tube-B:ripcord-1/2, and Triplex-2:tail-fiber. However, we left out the C12:C5 interface between portal and capsid, as the portal N-termini (res. 1-65) are not visible: these residues may play a crucial role in the interaction, so we refrain from speculating mismatched interfaces in the current paper.

4) The paper is rather thin in comparing structural findings with information available for other phage, in particular those belonging to Myoviridae (e.g. doi: 10.1038/emboj.2009.36; doi/10.1073/pnas.1605883113). Discussing structural data in a wider context is important for underlying significance of presented data.

The reviewer is correct, but the paper is ~1,200 words over the recommended length for a full article in Nature Communications. We do use Supplementary Figs. 3 and 6 to compare E217 proteins to other available phages. And we do cite the excellent papers mentioned by the reviewer: see references 16 and 39.

5) Most importantly, it would be useful to discuss the proposed mechanism “A pathway for signal transduction from tail fiber to sheath contraction” in comparison with mechanisms proposed earlier on the basis of structural data for other representatives of Myoviridae.

The reviewer is correct, but the paper is ~1,200 words over the recommended length for a full article in Nature Communications. We will refrain from adding more text.

I would abbreviate the Ramachandran plot statistics in Table 1 to the first decimal point (e.g. “7.1” instead of “7.11” etc).

Done!

Reviewer #3

This is a quality work showing the high resolution structures of E217 pseudomonas phage used in phage therapy. Structures were solved before and after tail contraction to release the genome at resolutions allowing 19 new proteins to be built into phage neck, sheath, and baseplate as well as tail fibers. The structures are impressive.

Thank you!

At the end of the first paragraph of Results, ...”obtained by combining different cryo EM maps.” What maps were combined to obtain what structures?

Done! See page 3, lines 146-147. “**e.g., (e.g., C5 capsid: EMD-29406; C6 neck: EMD-29487; C6 extended: EMD-29481; C6 contracted sheath: EMD-29486; C3 baseplate: EMD-28405).**”

By definition, virion is an infectious virus particle that contains a genome. Thus, the contracted form of E217 is not a virion.

Done! Virion was replaced with ‘**empty particles**’. See on page 3, lines 143, and page 15, line 607.

Supplemental Figure 2 may be important enough to bring into the paper as a Figure. The Atlas should include the pdb accession code with each protein. Figure 8 and 9 could move to supplemental.

Unfortunately, our paper exceeds the maximum number of figures/tables allowed in Nature Communications (10 panels). We have to leave Suppl. Figure 2 (which we agree is very informative) in the Supplementary Information.

All figures need more descriptive legends. The color coded key for Figures 2 and 6 is excellent, could it be used in other figures?

Done! Color-coded keys were also added to **Figures 1 and 7**.

The first two paragraphs of Discussion summarize the results without any real discussion.

Done! We moved this section to the Introduction (on page 1, lines 59-62): “**PB1-like phages are ubiquitous on earth⁵³, found in freshwater in the US⁵⁴ and Brazil⁵⁵, as well as sewage and wastewater in Europe⁵⁶. These phages have genomes of ~65 kbs and significantly simpler baseplate complexes than the classical coliphage T4**”